# Bridging the Translational Gap in Chemotherapy-Induced Peripheral Neuropathy with iPSC-Based Modeling

**DOI:** 10.3390/cancers14163939

**Published:** 2022-08-15

**Authors:** Christina Mortensen, Nanna Elman Andersen, Tore Bjerregaard Stage

**Affiliations:** 1Clinical Pharmacology, Pharmacy, and Environmental Medicine, Department of Public Health, University of Southern Denmark, DK-5000 Odense C, Denmark; 2Department of Clinical Pharmacology, Odense University Hospital, DK-5000 Odense C, Denmark

**Keywords:** chemotherapy-induced peripheral neuropathy, induced pluripotent stem cells, dorsal root ganglia, sensory neurons, Schwann cells, in vitro cell models, translational research

## Abstract

**Simple Summary:**

Chemotherapy-induced peripheral neuropathy (CIPN) remains a clinical challenge with a considerable impact on the effective treatment of cancers and quality of life during and after concluding chemotherapy. Given the limited understanding of CIPN, there are no options for the treatment and prevention of CIPN. Decades of research with the unsuccessful translation of preclinical findings to clinical studies argue for the requirement of human model systems. This review focuses on the translational potential of human induced pluripotent stem cells (iPSCs) in CIPN research. We provide an overview of the current studies and discuss important aspects to improve the translation of in vitro findings. We identified distinct effects on the neurite network and cell viability upon exposure to different classes of chemotherapy. Our study revealed considerable variability between donors and between neurons of the central and peripheral nervous system. Translational success may be improved by including multiple iPSC donors with known clinical data and selecting clinically relevant concentrations.

**Abstract:**

Chemotherapy-induced peripheral neuropathy (CIPN) is a common and potentially serious adverse effect of a wide range of chemotherapeutics. The lack of understanding of the molecular mechanisms underlying CIPN limits the efficacy of chemotherapy and development of therapeutics for treatment and prevention of CIPN. Human induced pluripotent stem cells (iPSCs) have become an important tool to generate the cell types associated with CIPN symptoms in cancer patients. We reviewed the literature for iPSC-derived models that assessed neurotoxicity among chemotherapeutics associated with CIPN. Furthermore, we discuss the gaps in our current knowledge and provide guidance for selecting clinically relevant concentrations of chemotherapy for in vitro studies. Studies in iPSC-derived neurons revealed differential sensitivity towards mechanistically diverse chemotherapeutics associated with CIPN. Additionally, the sensitivity to chemotherapy was determined by donor background and whether the neurons had a central or peripheral nervous system identity. We propose to utilize clinically relevant concentrations that reflect the free, unbound fraction of chemotherapeutics in plasma in future studies. In conclusion, iPSC-derived sensory neurons are a valuable model to assess CIPN; however, studies in Schwann cells and motor neurons are warranted. The inclusion of multiple iPSC donors and concentrations of chemotherapy known to be achievable in patients can potentially improve translational success.

## 1. Introduction

Chemotherapy-induced peripheral neuropathy (CIPN) remains a clinical challenge with considerable impacts on the efficacy of chemotherapy and quality of life during and after concluding chemotherapy. The number of cancer survivors is projected to increase substantially over the next decades due to advances in the early diagnosis and treatment of cancers. Consequently, the proportion of cancer patients suffering from CIPN are expected to increase in the coming decades [1]. According to clinical practice guidelines, there are no prevention and treatment options for CIPN. This leaves dose delaying, dose reduction or treatment discontinuation as countermeasures for patients with intolerable symptoms [2]. As a result, the efficacy of chemotherapy might be decreased and possibly have detrimental effects on clinical outcomes [3,4]. CIPN is caused by a substantial number of agents, including taxanes, vinca alkaloids, platinum-based agents and proteasome inhibitors. CIPN spans a broad spectrum of clinical symptoms depending on whether sensory, motor, or autonomic neurons are damaged. CIPN is primarily associated with sensory symptoms characterized by tingling, numbness, sensory abnormalities, and neuropathic pain in the extremities. Some agents also cause motor neuropathy (muscle weakness, muscle atrophy and cramping) and autonomic neuropathy (orthostatic hypotension, urinary retention, and erectile dysfunction). The mechanisms of each agent’s neurotoxicity, types of neurons affected and specific clinical symptoms have been described in detail elsewhere [5,6,7]. The heterogeneity of symptoms can be explained by classes of chemotherapeutic agents exhibiting different mechanisms of damage to the peripheral nervous system (PNS) [8]. However, variation in CIPN phenotypes may exist after treatment with the same agent [9,10].

Although the burden of CIPN is high, there is a lack of understanding of the molecular mechanisms underlying CIPN. Multiple mechanisms have been proposed, including microtubule disruption, mitochondrial dysfunction, neuroinflammation and altered ion channel dynamics [11]. The need for objective and standardized measures of CIPN created greater attention towards molecular predictors rather than conventional clinical predictors, such as age, sex and co-morbidity. The diagnosis and management of CIPN is challenging because its clinical presentation and disease course are heterogenous, and no clear consensus exists in international guidelines [12]. This highlights an urgent need to identify molecular biomarkers for discriminating CIPN phenotypes and assessing CIPN severity in research and clinical practice. Single nucleotide polymorphisms (SNPs) in genes encoding drug metabolizing enzymes and drug transporters have been shown to predict an individual patient’s risk of CIPN [13]. These genes are involved in regulating systemic drug concentrations (hepatic cytochrome P450) and neuronal intracellular drug concentrations (ABCC1 and ABCB1) [10]. Discordance across genome-wide association studies questions the utility of SNPs as a clinical tool to optimize and individualize chemotherapy treatment. Recently, neurofilament light chain has shown promise as a potential biomarker of neurotoxicity in iPSC-derived sensory neurons [14] and chemotherapy-induced peripheral neuropathy in cancer patients [15,16].

Despite decades of research into the prevention and treatment of CIPN, no therapeutic interventions have proven successful in clinical trials. This may suggest that human systems are required in the preclinical development of effective and safe interventions to reduce the burden of CIPN [17]. The discovery of induced pluripotent stem cells (iPSCs) from adult human cells enables the unlimited generation of almost any cell type of the human body. A huge effort has been made into establishing robust and reproducible differentiation protocols for several PNS cell types, including sensory neurons [18], Schwann cells [19] and motor neurons [20]. Previous studies indicated that iPSC-derived sensory neurons have shown to reflect the clinical features of CIPN, including degeneration of the distal nerve endings [21] and individual CIPN susceptibilities [22]. In this review, we provide an overview of studies utilizing human iPSC-derived models to study CIPN. We identify gaps in our current understanding of CIPN and provide guidelines for future translational research.

## 2. Methods

### 2.1. Search Strategy

We performed a systematic search using PubMed on 5 May 2021. An updated search was performed on 10 August 2022. We used a combination of MeSH terms, free-text terms and synonyms for chemotherapy-induced peripheral neuropathy, human induced pluripotent stem cells and PNS cell types. The search strategy is represented in Appendix A.

### 2.2. Selection Process

Titles and abstracts were independently screened for potential eligibility by two reviewers (C.M., and N.E.A.) using Covidence. After the initial screening, each reviewer individually assessed full-text articles for the eligibility criteria. We included original articles if they (i) employed human induced pluripotent stem-cell derived PNS cell types (sensory neurons, motor neurons, Schwann cells and satellite cells) and (ii) assessed CIPN. No additional studies were identified through review of reference lists for the included articles. An additional study was found by manual search. A flow diagram of the literature search is summarized in Appendix A. Any disagreements were resolved by discussion.

### 2.3. Data Extraction

The data were extracted into a structured table and included information about the cell model, applied chemotherapeutic agents and their respective concentration range, the half-maximal inhibitory concentration (IC_50_) values and analysis methods (Table 1). If IC_50_ values were not calculated in the respective articles, IC_50_ values were roughly estimated using concentration–response curves. Estimated IC_50_ values are indicated by ~ in Table 1. Data were extracted for CIPN- and non-CIPN-causing chemotherapeutic agents approved for cancer treatment by The United States Food and Drug Administration (FDA). The complete data extracted for iPSC-based studies are provided in a sortable Excel spreadsheet (Appendix A).

### 2.4. Estimation of Clinically Relevant Concentrations

PubMed was used to find clinical studies assessing the pharmacokinetics of widely used chemotherapeutic agents. We chose original articles and reviews with the most widely used dose regimen administered as a single dose. The maximum concentration (C_max_) values were extracted and converted into nano- or micromolar. Since the data presentation varied among studies, both median and mean estimates were extracted. Based on the plasma protein binding, the free concentration of the chemotherapeutic agents (C_max_ unbound) was estimated. Pharmacokinetic data of common paclitaxel and vincristine formulations are included in this compilation. For these, the C_max_ unbound was only reported if measured clinically using equilibrium dialysis or ultrafiltration.

## 3. Results

A total of 13 references were included in our review. The included studies utilized iPSC-derived neurons of either CNS or PNS identity. No studies with motor neurons, Schwann cells and satellite cells were found.

### 3.1. Neurotoxicity Assessment of Various Chemotherapeutic Agents Using iPSC-Derived PNS Models

We reviewed the literature for iPSC-derived models assessing neurotoxicity among different classes of chemotherapeutic agents associated with CIPN development. Current iPSC-derived models of CIPN are cultured in 2D and predominantly only include a single donor iPSC line. The most widely used differentiation approach is based on chemical treatment with small molecule inhibitors often followed by maturation with neurotrophic growth factors. However, the differentiation approach for commercially available iPSC-derived neurons is mostly unknown. Neurotoxicity is commonly evaluated using cell viability or neurite network analysis based on high-content imaging.

#### 3.1.1. Taxanes

The in vitro neurotoxicity of paclitaxel was reported in several studies using iPSC-derived neurons [21,22,25,26,27,28,29]. Paclitaxel and docetaxel reduce the complexity of the neuronal network with minimal or no effects on cell viability at clinically relevant concentrations (Table 1 and Table 2). A single study reported decreased cell viability of iPSC-derived sensory neurons (iPSC-SNs) after paclitaxel treatment. This can be explained by not including sufficiently high concentrations of paclitaxel to reach the maximum inhibitory effect (I_max_) [22]. iPSC-SNs and iPSC-derived peripheral-like neurons (Peri.4U) were sensitive towards paclitaxel and docetaxel as shown by IC_50_ values for neurite network analysis ranging from 0.7 nM to 1.4 µM [21,22,25,26,27,29]. iCell neurons that represent GABAergic and glutamatergic cortical neurons showed higher tolerance towards taxanes compared to peripheral neurons [21,22,23,25,26,27,28].

#### 3.1.2. Vinca Alkaloids

Vincristine decreased neurite network with low IC_50_ values of 5.5 nM to 0.4 µM across different iPSC-derived models [21,24,25,26,27,28]. Vincristine affected the cell viability at concentrations exceeding clinically relevant concentrations (Table 1 and Table 2).

#### 3.1.3. Platinum-Based Agents

Cisplatin and oxaliplatin caused minimal neurotoxicity to iPSC-derived neurons as indicated by IC_50_ values up to 100 µM for neurite network analysis. Prolonged treatment with these agents was necessary to observe an impact on neurite network and cell viability. The two platinum-agents reduced cell viability at concentrations higher than free drug concentration in patients. Carboplatin is not neurotoxic, which was verified by high IC_50_ values for both neurite network analysis and cell viability (Table 1).

#### 3.1.4. Proteasome Inhibitor

Bortezomib showed varying neurotoxic effects for the same cell model and across different cell models with IC_50_ values ranging from 3 nM to 100 µM. Considering the considerable variation between studies, bortezomib seems to affect both the neurite network and cell viability.

#### 3.1.5. Anti-Angiogenic Agents

Thalidomide, lenalidomide and pomalidomide showed no effects on the neurite network or cell viability. IC_50_ values were determined to be above 100 µM for both parameters, which is substantially higher than clinically relevant concentrations of thalidomide (Table 2) [24,26,27].

#### 3.1.6. Non-CIPN Causing Agents

5-Fluorouracil, hydroxyurea and doxorubicin are chemotherapeutic agents that do not cause CIPN. Therefore, these are often used as negative controls for in vitro experiments. The IC_50_ values for 5-fluorouracil and hydroxyurea could not be determined after exposure to the highest concentrations (10 µM and 100 µM) by neurite network analysis or cell viability [22,27]. However, the applied concentration of 5-fluorouracil and hydroxyurea did not reach clinically relevant concentrations of 384 and 595 µM respectively. Due to the high-dose and low binding to plasma proteins, the clinically relevant concentrations of these agents are higher than the other chemotherapeutic agents. One study estimated doxorubicin IC_50_ to 408.8 µM [22], which is ~70-fold higher than the free drug concentration observed in humans (Table 2).

### 3.2. Differential Chemotherapy Susceptibility among Donors

Two studies aimed to assess chemotherapy susceptibility using iPSC-derived neurons from different healthy donors. IC_50_ values for vincristine, cisplatin, bortezomib and paclitaxel varied substantially between donors, indicating their individual tolerance to neurotoxic chemotherapy (Table 3). Since there is no clinical data for the included donors, it is not known whether the donor-to-donor variability is due to age, sex, disease, ancestry or genetic factors.

### 3.3. Identification of Clinically Relevant Concentrations for In Vitro Studies

We generated a list of commonly used chemotherapeutic agents with the standard dose, maximum plasma concentration, plasma protein binding and the estimated free drug concentration for in vitro studies (Table 2). The fraction of chemotherapy agent bound to plasma proteins varies widely between chemotherapeutic agents, and this significantly impacts available chemotherapy drug in the plasma. The free drug concentration is also modified by the formulation of the agent, and thus we included two standard formulations of paclitaxel and vincristine. The raw C_max_ values as extracted from the literature are included in Appendix A.

## 4. Discussion

In this review, we summarized the utility of iPSC-derived neurons in CIPN research. We identified distinct effects on the neurite network and cell viability upon exposure to different classes of chemotherapy. The iPSC-derived neurons are a highly relevant model to assess specific molecular mechanisms underlying CIPN phenotypes. There is considerable donor-to-donor variability that needs to be considered when interpreting and comparing findings. The reviewed studies often include chemotherapy concentrations greater than what can be achieved in a clinical setting. A fundamental aspect for in vitro to in vivo translation is the use of chemotherapy concentrations known to be achievable in cancer patients. To aid in study design for future translational CIPN research, we provided pharmacokinetic estimates for clinically relevant concentrations of approved and widely used chemotherapeutic agents.

While paclitaxel, docetaxel, vincristine and bortezomib exhibited substantial neurotoxicity, minimal neurotoxicity was observed in iPSC-derived neurons after treatment with cisplatin and oxaliplatin. This might indicate that single exposure of these platinum-based agents is associated with functional impairment rather than structural damage. Another explanation could be that the initial steps in the cascade responsible for oxaliplatin-induced peripheral neuropathy occur in another PNS cell type, such as Schwann cells.

Oxaliplatin affects specific ion channels on peripheral neurons leading to cold hypersensitivity, a unique feature of oxaliplatin-induced peripheral neuropathy [46]. Platinum-based agents have a delayed onset of symptoms that often worsen after treatment cessation, a phenomenon called coasting [47]. Based on these clinical observations, prolonged treatment for up to 96 hr or repeated exposure to platinum-based agents might better reflect the phenotype observed in cancer patients. However, it is also possible that researchers should look for other phenotypes.

CIPN has mostly been assessed in vitro using commercially available neurons derived from iPSCs (iCell neurons and Peri.4U neurons). iCell neurons are immature forebrain neurons, whereas Peri.4U neurons are considered peripheral-like neurons. The cells are used for experiments shortly after seeding, and thereby they are not allowed to mature and establish a comprehensive neuronal network. iPSC-SNs display the characteristically dorsal root ganglia (DRG) morphology where cell bodies are organized in large ganglia with numerous of emanating axons.

Contrary to commercially available iPSC-derived neurons, iPSC-SNs express canonical DRG markers, such as receptors involved in the perception of pain, including TRPV1, TRPM8 and PIEZO2 [21,22]. The ideal model to study CIPN should also reflect the heterogeneity of sensory neuron subtypes. Early studies verified that iPSC-SNs contain multiple sensory neuron subtypes, including peptidergic and non-peptidergic nociceptors, mechanoreceptors and proprioceptors [21,22,24]. The contribution of individual sensory neuron subtypes to the molecular phenotype of CIPN is poorly understood. Advanced and hypothesis-free approaches, such as single-cell transcriptomics and single-cell proteomics, are powerful tools to reveal the sensory neuron subtype-specific molecular characteristics of CIPN across various chemotherapeutic agents. This might enable researchers to identity common molecular targets across chemotherapeutics agents, which would be valuable regarding the development of new therapeutics for CIPN.

Patient-specific iPSCs have considerable advantages over previously used models. They are expected to reproduce human pathophysiology and recapitulate the considerable interindividual variation that exists within patient populations, which may play a role in chemotherapy tolerance and CIPN severity. Patient-specific iPSCs maintain the genotype encoded by the donor; this enables researchers to understand the mechanisms underlying CIPN at the individual patient level. The establishment of patient-specific iPSC-SNs for groups of patients with severe CIPN and groups of patients tolerating chemotherapy with asymptomatic or mild CIPN might reveal new mechanistic insights into CIPN pathogenesis. Such an approach might identify molecular predictors than can subsequently be validated in cancer patients aiming to stratify low- and high-risk groups for CIPN or developing novel therapies for CIPN.

iPSC-SNs have been utilized to prospectively predict drug response in a patient with small fiber neuropathy [48]. They demonstrated that lacosamide could reduce excitability in iPSC-SNs, and this correlated to microneurography recordings of the patient’s nerve fibers and alleviation of neuropathic pain reported by the patient. Although this is a proof-of-principle from a single study, iPSC-SNs might be a powerful tool to predict patient-specific drug responses for various pain syndromes, including CIPN.

Furthermore, patient-specific iPSC-SNs can be used to study genotype-phenotype associations of SNPs within genes related to sensory neurons known or suspected to be associated with an increased risk of CIPN. As all other models, the iPSC-SN model has limitations. The main limitation is the amount of time required for reprogramming and validating iPSCs and the subsequent differentiation of these into sensory neurons (≈5–6 months). Another limitation is that reprogramming resets the epigenetic state of somatic cells and erases their age of origin. In addition, a large-scale study (123 iPSC-SN differentiations) showed batch-to-batch variability and the presence of non-neuronal cell types, including fibroblasts [49]. Finally, iPSC-SNs are sensitive and can easily detach, thereby, requiring a certain level of expertise.

The primary weakness of this review is that we only focused on structural damage and alterations in cell viability as indicators of neurotoxicity. Some chemotherapeutic agents might cause acute peripheral neuropathy by affecting the function of sensory neurons independently of structural damage, such as oxaliplatin and cisplatin. However, the functional impairment of sensory neurons caused by chemotherapy has scarcely been addressed and is not included in this review. Additionally, different methods to analyze the neurite network and cell viability have been utilized across studies. While different methods might lead to variation in the estimated IC_50_ values, we did not distinguish between these when extracting the data. The number of publications within this field is sparse, and thus we only collected data from a total of 13 references. Another limitation of our study is that we extracted the mean and median C_max_ values from a single study. Consequently, the estimated concentrations for in vitro studies does not directly reflect the interindividual variability.

We found that many studies are limited by including only a single iPSC donor. The clinical presentation and disease course of CIPN are heterogenous, and thus the inclusion of multiple iPSC donors is encouraged to ensure the generalizability of in vitro findings. Donors have distinct and typically unknown genetic backgrounds that influence the resulting phenotype, i.e., whether the donor is chemotherapy-tolerant (low-risk CIPN neurons) or chemotherapy-sensitive (high-risk CIPN neurons). Thus, the genetic background of donors might easily confound end-point measurements, such as morphological or functional changes. Therefore, the ideal study design should include multiple donors and preferably both male and female donors to account for interindividual variability.

The included studies utilized different approaches to calculate IC_50_ values, which may partially explain the large variability in the estimated IC_50_ values among different studies assessing the same agent. The level of neurotoxicity can easily be over- or underestimated if the IC_50_ values are estimated based on concentration–response curves not covering the whole concentration range. Another important limitation of the current literature is the lack of the formulation of chemotherapy used clinically. One study explored the difference in neurotoxicity of nab-paclitaxel and paclitaxel. However, the in vitro bioequivalence of the formulations was not established. In vitro bioequivalence is achieved if the free, unbound fraction of chemotherapy in the culture medium is equal between formulations. The free fraction of chemotherapeutic agents represents the pharmacologically active chemotherapeutic agents that can interact with molecular targets and induce neurotoxicity. The traditional formulation of paclitaxel (Taxol) is based on polyoxyethylated castor oil (Cremophor-EL, CreEL), which is known to reduce the free, unbound fraction available for drug-target interactions due to the entrapment of paclitaxel into CreEL micelles [50]. Additionally, CreEL has shown to be neurotoxic on its own, and thus dissolving paclitaxel and other agents in their clinically used formulations may improve translation.

While multiple cell types of the PNS have been proposed to be involved in CIPN pathogenesis, only human iPSC-derived sensory neurons have been used to model CIPN. To our knowledge, Schwann cells and motor neurons of human origin have not yet been utilized to understand CIPN. The currently used 2D iPSC-SN models contain unmyelinated fibers, and thus co-culture or organoid systems containing Schwann cells are warranted. The prevalence of motor symptoms among patients treated with vincristine and paclitaxel underlines the importance of utilizing motor neurons to study CIPN [51,52].

We believe that the ideal PNS model recapitulates the in vivo intercellular heterogeneity and reflects the in vivo architecture. The creation of a 3D system containing multiple iPSC-derived PNS cell types would likely improve physiological mimicry and clinical translation. The field of 3D PNS modeling is constantly growing. A few 3D models have been developed to study PNS diseases that mimic the physiological environment more closely. One study co-cultured human iPSC-derived sensory neurons with primary rat Schwann cells using a scaffold platform with hydrogel embedding [53]. Another study cultured spheroids consisting of iPSC-derived motor neurons and primary human Schwann cells on a nerve-on-a-chip platform [54]. These models recapitulate PNS complexity and neuron-glia interplay, which is necessary for understanding the complex pathophysiology of CIPN in humans. The inclusion of not only Schwann cells but also satellite glial cells and immune cells is crucial since they can produce pro-inflammatory cytokines, sensitize sensory neurons and contribute to the onset and maintenance of CIPN.

The production of cytokines might also be augmented by the secretion of various neuropeptides from the sensory neurons, including substance P and CGRP. A recent study also found that the secretion of neuropeptides from the sensory neurons that innervates the lung promoted the severity of COVID-19. This, in turn, may highlight the importance of assessing chemotherapy-induced inflammation and its regulation of pain [55].

## 5. Conclusions

In conclusion, the current literature regarding iPSC-derived cell models indicates that mechanistically diverse chemotherapeutics are neurotoxic to varying degrees. We found substantial variation between different cell models assessing the same agent, indicating that the neuron identity and donor background need to be considered when studying CIPN. Several studies utilized concentrations of chemotherapy greater than what can be achieved in patients. To improve clinical translation, we suggest that clinically relevant concentrations are based on the free, unbound fraction of chemotherapeutics or that chemotherapeutics are used in formulations that correspond to those used clinically. Since multiple cell types exist in the PNS, studies in Schwann cells, motor neurons and co-culture systems are warranted to fill gaps in our current knowledge of CIPN. Human iPSC-based models allow for detailed studies of the molecular mechanisms underlying CIPN and may provide a translational bridge from in vitro findings to clinical studies with the aim of reducing the burden of CIPN.

## Figures and Tables

**Table 1 cancers-14-03939-t001:** Comparison of IC_50_ values across different models of human induced pluripotent stem cell-derived neurons and across various chemotherapeutic agents and their applied concentrations and duration of chemotherapy exposure.

Cell Type	Number of Donors	Chemotherapeutic Agents	Concentrations Applied	IC_50_		Analysis	Reference
				Neurite Network Analysis	Cell Viability		
Human iPSC-derived sensory neurons (iPSC-SNs)	4–6	Paclitaxel	0.1–1 µM	0.1 µM (48 h)	>1 µM (48 h)	Single-cell sequencing, neurite network analysis, cell viability, siRNA transfection, electrophysiology	[23]
Human iPSC-derived peripheralneurons	1	Vincristine	1 nM–100 µM	~0.4 µM (24 h)	~75 µM (24 h)	Cell viability, apoptosis, neurite network analysis, qRT-PCR, ELISA	[24]
		Ixabepilone	1 nM–100 µM	~0.45 µM (24 h)	>100 µM (24 h)		
		Cisplatin	1 nM–100 µM	~65 µM (24 h)	~100 µM (24 h)		
		Bortezomib	1 nM–100 µM	~100 µM (24 h)	>100 µM (24 h)		
		Pomalidomide	1 nM–100 µM	~95 µM (24 h)	>100 µM (24 h)		
iPSC-SNs	2	Paclitaxel	100 pM–10 µM	-	290 nM (24 h) 99.1 nM (72 h)	Bulk RNA sequencing, electrophysiology, calcium imaging, immunocytochemistry (ICC), bright field microscopy, cell viability	[22]
		Vincristine	100 pM–10 µM	-	66.3 nM (24 h)		
		Cisplatin	1 nM–100 µM	-	11.7 µM (24 h)		
		Bortezomib	100 pM–10 µM	-	3.8 nM (24 h)		
		Negative controls:					
		Doxorubicin	100 pM–10 µM	-	408.8 µM (24 h)		
		5-Fluorouracil	100 pM–10 µM	-	>10 µM (24 h, 72 h)		
iPSC-SNs	1	Paclitaxel	1 nM–50 µM	1.4 µM (48 h) 0.6 µM (72 h)	38.1 µM (48 h) 9.3 µM (72 h)	Neurite network analysis, qRT-PCR, calcium imaging, viability, apoptosis, mitochondrial measurements	[21]
		Docetaxel	0.1–1 µM	~1 µM (72 h)	-		
		Vincristine	10 nM–1 µM	~0.01 µM (72 h)	-		
		Bortezomib	0.1–1 µM	~1 µM (72 h)	-		
		Negative control:					
		Hydroxyurea	0.1–1 µM	>1 µM (72 h)	-		
iPSC-SNs	2	Paclitaxel	10 nM–10 µM	5 nM (48 h)	7.4 µM (48 h)	Neurite network analysis, calcium imaging, cell viability	[25]
		Vincristine	10 nM–10 µM	63 nM (48 h)	0.6 µM (48 h)		
		Cisplatin	10 nM–10 µM	5 nM (48 h)	3.1 µM (48 h)		
		Bortezomib	10 nM–10 µM	4.2 µM (48 h)	1 µM (48 h)		
		Etoposide	10 nM–10 µM	19 nM (48 h)	3.2 µM (48 h)		
Peri.4U neurons	1	Paclitaxel	130 pM–10 µM	5.6 nM (24 h)	>10 µM (24 h)	Electrophysiology, cell viability, ICC	[26]
		Docetaxel	130 pM–10 µM	0.7 nM (24 h)	>10 µM (24 h)		
		Vincristine	10 pM–1 µM	5.5 nM (24 h)	>1 µM (24 h)		
		Ixabepilone	130 pM–10 µM	4.3 nM (24 h)	>10 µM (24 h)		
		Cisplatin	1.3 nM–100 µM	>100 µM (24 h)	>100 µM (24 h)		
		Oxaliplatin	1.3 nM–100 µM	74.1 µM (24 h)	>100 µM (24 h)		
		Carboplatin	1.3 nM–100 µM	>100 µM (24 h)	>100 µM (24 h)		
		Bortezomib	1.3 nM–100 µM	>100 µM (24 h)	>100 µM (24 h)		
		Thalidomide	1.3 nM–100 µM	>100 µM (24 h)	>100 µM (24 h)		
		Lenalidomide	1.3 nM–100 µM	>100 µM (24 h)	>100 µM (24 h)		
		Pomalidomide	1.3 nM–100 µM	>100 µM (24 h)	>100 µM (24 h)		
		Negative control:					
		Hydroxyurea	1.3 nM–100 µM	>100 µM (24 h)	>100 µM (24 h)		
Commercial human iPSC-derived neurons (iCell)Peri.4U neurons	1	Paclitaxel	iCell neurons: 1 nM–100 µMPeri.4U neurons: 0.01 nM–100 µM	iCell neurons:		Electrophysiology, neurite network analysis, cell viability, apoptosis	[27]
				~10 µM (72 h)	~100 µM (72 h)		
				Peri.4U neurons:			
				~1 µM (72 h)	>10 µM (72 h)		
		Nab-paclitaxel		iCell neurons:			
				~10 µM (72 h)	>100 µM (72 h)		
		Docetaxel		iCell neurons:			
				~10 µM (72 h)	>100 µM (72 h)		
		Vincristine		iCell neurons:			
				~0.1 µM (72 h)	>10 µM (72 h)		
				Peri.4U neurons:			
				<1 nM (72 h)	~40 nM (72 h)		
		Cisplatin		iCell neurons:			
				~10 µM (72 h)	~7.94 µM (72 h)		
				Peri.4U neurons:			
				~7.94 µM (72 h)	~3.16 µM (72 h)		
		Oxaliplatin		iCell neurons:			
				~31.6 µM (72 h)	~20 µM (72 h)		
		Carboplatin		iCell neurons:			
				~63.1 µM (72 h)	~39.8 µM (72 h)		
		Bortezomib		iCell neurons:			
				~32 nM (72 h)	~6 nM (72 h)		
				Peri.4U neurons:			
				~3 nM (72 h)	~40 nM (72 h)		
		Thalidomide		iCell neurons:			
				>100 µM (72 h)	>100 µM (72 h)		
		Negative controls:					
		5-Fluorouracil		iCell neurons:			
				>100 µM (72 h)	>100 µM (72 h)		
iCell neurons	1	Paclitaxel	1 nM–100 µM	~20 µM (72 h)	-	Transfection, cell viability and apoptosis, neurite network analysis, time-lapse microscopy	[28]
		Vincristine	1 nM–100 µM	~0.04 µM (72 h)	-		
		Cisplatin	1 nM–100 µM	~10 µM (72 h)	-		
		Negative controls:					
		Hydroxyurea	0.001–100 µM	>100 µM (72 h)	-		

**Table 2 cancers-14-03939-t002:** Overview of key human pharmacokinetic parameters for chemotherapeutics that cause CIPN and relevant negative controls.

Generic Name	Brand Name	Dose	Route	Infusion	Maximum Plasma Concentration	Plasma Protein Binding	Free Drug in Plasma	Reference
**Taxanes**								
Paclitaxel	Taxol	175 mg/m^2^	IV	3 h	5.1 µM ^+^	89–98%	0.1–0.4 µM *	[30,31]
Nab-paclitaxel	Abraxane	260 mg/m^2^	IV	1 h	9.5 µM ^#^		0.5 µM *	[32]
Docetaxel	Taxotere	100 mg/m^2^	IV	1 h	5.1 µM ^#^	98%	0.1 µM	[33]
Ixabepilone	Ixempra	30 mg/m^2^	IV	1 h	0.7 µM	67–77%	0.2 µM	[34]
**Vinca alkaloids**								
Vincristine	Vincasar PFS	1.5–2 mg/m^2^	IV	Bolus or 1 h	36–88 nM ^+^	75%	9.1–22 nM	[35]
Vincristine (liposomal)	Marqibo	2.25 mg/m^2^	IV	1 h	2.6 µM ^+^		n.d.	[36]
**Platinum-based agents**								
Cisplatin	Platinol	80 mg/m^2^	IV	2 h	11 µM ^#^	90%	1.1 µM	[37]
Oxaliplatin	Eloxatin	130 mg/m^2^	IV	2 h	6.5–8.1 µM ^#^	90%	0.7–0.8 µM	[38]
Carboplatin	Paraplatin	400 mg/m^2^	IV	0.5 h	134.7 µM ^#^	0%	134.7 µM	[39]
**Proteasome inhibitors**								
Bortezomib	Velcade	1.3 mg/m^2^	IV	Bolus	0.3 µM ^#^	83%	53.1 nM	[40]
**Anti-angiogenic agents**								
Thalidomide	Thalomid	200 mg	PO	-	3.9–7.7 µM	55% (R)-(+) 65% (S)-(−)	1.7–3.5 µM 1.4–2.7 µM	[41]
**Non-CIPN causing agents (negative controls)**								
5-Fluorouracil	Adrucil	400 mg/m^2^	IV	Bolus	426 µM ^+^	10%	383.6 µM	[42]
Hydroxyurea	Droxia	2000 mg	PO	-	794 µM ^#^	25%	595 µM	[43,44]
Doxorubicin (liposomal)	Caelyx	30 mg/m^2^	IV	1 h	18.3 µM ^#^	70%	5.5 µM	[45]

Note: * refers to measured free drug concentrations in human plasma, ^+^ refers to median and ^#^ refers to mean. Abbreviations: hr, hours; IP, intraperitoneal; IV, intravenous; PO, per os; n.d., not determined.

**Table 3 cancers-14-03939-t003:** Donor backgrounds affect the half-maximal inhibitory concentration (IC_50_) of various chemotherapeutic agents in human induced pluripotent stem cell-derived neurons.

Model	Subject	Chemotherapeutic Agents	Concentrations Applied	IC_50_		Reference
				Neurite Network Analysis	Cell Viability	
iPSC-SNs	Donor 1	Vincristine	100 pM–10 µM	-	87.6 nM (24 h)	[22]
		Cisplatin	1 nM–100 µM	-	14.7 µM (24 h)	
		Bortezomib	100 pM–0.1 µM	-	5.3 nM (24 h)	
	Donor 2	Vincristine	100 pM–10 µM	-	35.8 nM (24 h)	
		Cisplatin	1 nM–100 µM	-	6.9 µM (24 h)	
		Bortezomib	100 pM–0.1 µM	-	1.3 nM (24 h)	
Human iPSC-derived neurons (MyCell)	Donor 1	Paclitaxel	0.001–100 µM	0.7 µM (72 h)	-	[28]
	Donor 2			60 µM (72 h)	-	
	Donor 3			2 µM (72 h)	-	
	Donor 4			15 µM (72 h)	-	

## Data Availability

Data are available in the Appendix A.

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
