# Peer review of "Bridging the Translational Gap in Chemotherapy-Induced Peripheral Neuropathy with iPSC-Based Modeling"

_cancers, 2022, doi:10.3390/cancers14163939_

Round 1

Reviewer 1 Report

This is a well-written and state-of-art review focusing on the utility of iPSC-based modeling in CIPN research. The review covers commonly used chemotherapeutic agents and identified distinct effects on neurite network and cell viability upon exposure to different chemotherapies. Several concerns should be addressed:

1. The authors did not explain why they only searched on "Pubmed". In addition, the searching data should be updated.

2. A total of 12 references and the sample size of each group were relevantly small.

3. Neurotoxicity assessment indicators of various chemotherapeutic agents are not clear enough. There are many ways to estimate cell viability or neurite network.

4. Different types of chemotherapy drugs have different neurotoxicity to (same/different) cells. Therefore, the side effects and the types of damage of chemotherapy drugs should be described in detail.

5. In the "Discussion" section, bridging the translational gap from in vitro findings to clinical studies needs deeper discussion.

6. The line 208 "discussion" should be in line 217. Additionally, the spell of "discussion" was wrong.

Author Response

This is a well-written and state-of-art review focusing on the utility of iPSC-based modeling in CIPN research. The review covers commonly used chemotherapeutic agents and identified distinct effects on neurite network and cell viability upon exposure to different chemotherapies. Several concerns should be addressed:

  1. The authors did not explain why they only searched on "Pubmed". In addition, the searching data should be updated.

Answer: We did not expect to find other relevant articles in other databases and thus, we performed the literature search in PubMed only. We agree with the reviewer’s comment that the literature search should be updated to ensure that newly published articles are also included.

Implemented change to manuscript:

Section 2.1, page 3, line 102-103: “An updated search was performed on August 10, 2022”

  1. A total of 12 references and the sample size of each group were relevantly small.

Answer: This is true. We have now mentioned it as a limitation of the review.

Implemented change to manuscript:

Section 4, page 11, lines 346-347: “The number of publications within this field is sparse and thus, we only collected data from a total of 13 references.”

  1. Neurotoxicity assessment indicators of various chemotherapeutic agents are not clear enough. There are many ways to estimate cell viability or neurite network.

Answer: We understand your concern. While our data indicate inter-laboratory variation in estimated IC50 values, this might partially be explained by the utilization of different methods to analyze neurite network and cell viability. Other factors may also contribute to this variation, such as expertise in differentiating and handling iPSC-derived neurons and different approaches to calculate IC50 values. We have mentioned this a limitation of our review instead of summarizing the utilized methods for each study.

Implemented change to manuscript:

Section 4, page 11, lines 343-346 “Additionally, different methods to analyze neurite network and cell viability have been utilized across studies. While different methods might lead to variation in estimated IC50 values, we did not distinguish between these when extracting the data.”

  1. Different types of chemotherapy drugs have different neurotoxicity to (same/different) cells. Therefore, the side effects and the types of damage of chemotherapy drugs should be described in detail.

Answer: We acknowledge your input; however, this is not within the scope of this review and have already been described previously.

Implemented change to manuscript:

Section 1, page 2, lines 58-60: “The mechanisms of each agent’s neurotoxicity, types of neurons affected, and specific clinical symptoms have been described in detail elsewhere [5–7].”

  1. In the "Discussion" section, bridging the translational gap from in vitro findings to clinical studies needs deeper discussion.

Answer: Thank you for your input on the discussion. Although a little unspecific, we agree that the discussion needed additional examples of how iPSC-derived sensory neurons can help to bridge the gaps currently present. The key points mentioned in the discussion should also be seen a best practice for how to improve translation of in vitro findings to the clinic.  

Implemented change to manuscript:

Section 4, page 11, lines 312-330: “Patient-specific iPSCs have considerable advantages over previously used models. They are expected to reproduce human pathophysiology and recapitulate the considerable interindividual variation that exists within patient populations, which may play a role for chemotherapy tolerance, and CIPN severity. Patient-specific iPSCs maintain the genotype encoded by the donor; this enables researchers to understand mechanisms underlying CIPN at an individual patient level. Establishment of patient-specific iPSC-SNs for groups of patients with severe CIPN and groups of patients tolerating chemotherapy with asymptomatic or mild CIPN might reveal new mechanistic insights into CIPN pathogenesis. Such an approach might identify molecular predictors than can subsequently be validated in cancer patients aiming to stratify low- and high-risk groups for CIPN or developing novel therapies for CIPN. iPSC-SNs have been utilized to prospectively predict drug response in a patient with small fiber neuropathy [48]. They demonstrated that lacosamide could reduce excitability in iPSC-SNs, and this correlated to microneurography recordings of the patient’s nerve fibers and alleviation of neuropathic pain reported by the patient. Although this is a proof-of-principle from a single study, iPSC-SNs might be powerful tool to predict patient-specific drug responses for various pain syndromes, including CIPN. Furthermore, patient-specific iPSC-SNs can be used to study genotype-phenotype associations of SNPs within genes related to sensory neurons known or suspected to be associated with an increased risk of CIPN.”

  1. The line 208 "discussion" should be in line 217. Additionally, the spell of "discussion" was wrong.

Answer: Thank you for mentioning this. We have now moved section 3.3 and corrected the misspelling.

Reviewer 2 Report

In this manuscript, the authors have reviewed Chemotherapy-induced peripheral neuropathy (CIPN) and present valuable insights and thoughts, which may have practical utility in reducing the burden of CIPN in cancer patients.

The review is timely and may be helpful to the clinicians and the basic researchers of cancer drug discovery.

Author Response

In this manuscript, the authors have reviewed Chemotherapy-induced peripheral neuropathy (CIPN) and present valuable insights and thoughts, which may have practical utility in reducing the burden of CIPN in cancer patients.

The review is timely and may be helpful to the clinicians and the basic researchers of cancer drug discovery.

Answer: We would like to thank and acknowledge Reviewer 2 for the positive feedback on this manuscript.